# Early Corticosteroid Therapy for *Mycoplasma pneumoniae* Pneumonia Irrespective of Used Antibiotics in Children

**DOI:** 10.3390/jcm8050726

**Published:** 2019-05-22

**Authors:** Eun-Ae Yang, Hyun-Mi Kang, Jung-Woo Rhim, Jin-Han Kang, Kyung-Yil Lee

**Affiliations:** 1Department of Pediatrics, College of Medicine, The Catholic University of Korea, Seoul 06591, Korea; anni79@catholic.ac.kr (E.-A.Y.); pedhmk@gmail.com (H.-M.K.); jwrhim@catholic.ac.kr (J.-W.R.); kjhan@catholic.ac.kr (J.-H.K.); 2Department of Pediatrics, The Catholic University of Korea Daejeon St. Mary’s Hospital, Daejeon 34943, Korea

**Keywords:** *Mycoplasma pneumoniae* pneumonia, macrolide antibiotics, antibiotic resistance, corticosteroids, prednisolone, methylprednisolone, children

## Abstract

Antibiotics’ effect on *Mycoplasma pneumoniae* (MP) infection still remains controversial. A prospective study of 257 children with MP pneumonia during a recent epidemic (2015–2016) was conducted. All MP pneumonia patients were treated with corticosteroids within 24–36 h after admission. Initially, oral prednisolone (1 mg/kg) or intravenous methylprednisolone (IVMP; 1–2 mg/kg) was administered for mild pneumonia patients, and IVMP (5–10 mg/kg/day) for severe pneumonia patients. If patients showed a persistent fever for 36–48 h or disease progression, additive IVMP (5 mg/kg or 10 mg/kg) was given. Thirty-three percent of patients received only a broad-spectrum antibiotic without a macrolide. The mean age and the male-to-female ratio was 5.6 ± 3.1 years and 1:1, respectively. Seventy-four percent of patients showed immediate defervescence within 24 h, and 96% of patients showed defervescence within 72 h with improvements in clinical symptoms. Three percent of patients (8/257) who received additive IVMP also showed clinical improvement within 48 h without adverse reactions. There were no clinical or laboratory differences between patients treated with a macrolide (*n* = 172) and without (*n* = 85). Early corticosteroid therapy might reduce disease morbidity and prevent disease progression in MP pneumonia patients without side effects, and antibiotics may have limited effects on MP infection.

## 1. Introduction

*Mycoplasma pneumoniae* (MP) is one of major respiratory pathogens that cause community-acquired pneumonia affecting children and young adults around the world [1]. Nationwide MP pneumonia epidemics have occurred with 3- to 4-year cycles in South Korea during recent decades [2,3].

MP has been regarded as a small extracellular bacterium that is highly sensitive to susceptible antibiotics, including macrolides, tetracyclines, and quinolones in vitro [4]. Although earlier a few case-series studies reported that erythromycin and tetracycline were effective against MP infection [5,6], there have existed some patients who have progressive severe pneumonia, which is not responsive to susceptible antibiotics in MP pneumonia epidemics [7,8]. For these patients, many investigators have reported that corticosteroids effectively initiate the rapid improvement of clinical symptoms and chest radiographic findings in children and adults with macrolide-sensitive MP (MSMP) or macrolide-resistant MP (MRMP) pneumonia [8,9,10]. 

The immunopathogenesis of MP pneumonia remains unknown, but it is believed that excessive immune reaction against the insults from MP infection is associated with lung cell injury [11]. With these clinical findings, epidemiological characteristics of MP infection, such as cyclic epidemics and case-predominance in young children, are similar to those of respiratory viral infections such as measles during the pre-vaccine era [1]. Therefore, there has been a long-standing controversy regarding the effects of antibiotics on MP infection in children [12,13].

MRMP strains have recently become prevalent, comprising over 80%–90% of cases in East Asian countries such as Japan and China [14,15]. Research groups in Korea have reported that 63% of isolated MP strains were MRMP strains in 2011 and 87.2% in 2015 nationwide epidemics [16,17]. Although some investigators reported that patients with MRMP pneumonia have prolonged fever duration and more severe morbidity compared to patients with MSMP pneumonia [18,19,20], true treatment failure of macrolides in MRMP pneumonia is very rare. 

We have reported the effectiveness of corticosteroids for the treatment of antibiotic non-responsive MP pneumonia since the nationwide MP epidemics in 2003, and found that earlier corticoid treatment is more effective to reduce MP pneumonia morbidity [8,21,22]. During the recent 2015–2016) epidemic in Korea, we had a plan to use early corticosteroids for all MP pneumonia patients, and the dose of corticosteroids was decided according to the severity of disease. Since the main MP strains in this epidemic were suspected to be MRMP, we had a chance to evaluate the effect of antibiotics on MP infection. In the present study, we aimed to present again the corticosteroid therapy as one of immune modulators with no treatment failures and discuss on limitations of antibiotic treatment for MP infection.

## 2. Patients and Methods 

### 2.1. Patient Selection

The study population included patients who were diagnosed with MP pneumonia (*n* = 257) at the Catholic University of Korea Daejeon St. Mary’s Hospital between January 2015 and December 2016. Diagnoses of MP pneumonia were made when patients showed clinical and serologic evidence of MP pneumonia. All patients had clinical evidence of pneumonia such as a fever, cough, and/or respiratory distress, and pneumonic infiltrations in the chest radiograph. Included patients underwent anti-MP IgM titrations twice by the micro-particle agglutination method (Serodia-Myco II, Fujirebio, Japan, positive ≥1:40), at the time of admission and before discharge. Patients were selected when they showed a seroconversion (negative to positive), four-fold or greater increase in IgM titers, or both high titers of ≥1:640 [23]. In this series, serum titration of IgM was performed to 1:1280, and titers over 1:1280 were reported as ≥1280. All subjects were previously healthy and lacked family histories of chronic lung diseases such as tuberculosis. Exclusion criteria were as follows: Patients who were examined by single testing, those who did not exhibit increased or decreased titers at the second test (≤1:320 at first examination) as false positives, who lacked fever or infiltration in the chest radiographs at presentation, who did not receive corticosteroids, or who had chronic disease states predisposing them to recurrent lung infections such as severe cerebral palsy or immunodeficiency. 

### 2.2. Corticosteroid and Antibiotic Treatment 

All patients were treated with corticosteroids within 24–36 h after admission. Initially, patients received oral prednisolone (1 mg/kg, divided into three per day) or low doses of IVMP (1–2 mg/kg, divided into two doses per day) if they had milder pneumonia lesions, and high-dose IVMP (5 mg/kg/day or 10 mg/kg/day) if they had more severe segmental/lobar lesions or severe respiratory distress such as wheezing or tachypnea needed oxygen supply, with or without extensive lung lesions at presentation. When patients exhibited persistent fever for 36–48 h after initial steroid therapy or signs of disease progression, additive high-dose MP (5 mg/kg/day or 10 mg/kg/day) was infused. Initial doses of oral prednisolone and low-dose IVMP were maintained for 2–3 days and tapered over one week, while high-dose IVMP was tapered to a half-dose on a daily basis if defervescence was noted; in these patients, therapy was changed to oral prednisolone and tapered over a period of one week. 

We hypothesized that antibiotics play a limited role in the pathogenesis of acute lung injury in MP infection through our experiences obtained from the previous epidemics. In this study, we prospectively planned to use one broad-spectrum and macrolide antibiotic for approximately half of the patients, and without macrolide for the other half. However, the numbers of cases of each group were slightly different. All patients were treated with a beta-lactam antibiotic (cefuroxime or amoxicillin/clavulanate), and two-thirds of patients received additional clarithromycin (*n* = 172), and one-third of patients were treated with a beta-lactam antibiotic only (*n* = 85). Pneumonic infiltration in the chest radiography was divided into two patterns, bronchopneumonia and segmental/lobar pneumonia (reviewed by Drs. EA Yang and KY Lee). The former was characterized by increased peribronchial or interstitial densities in one or both lung fields, while the latter was characterized by clear increased segmental or lobar filtration or consolidation in one or both lung fields as used in previous studies [21,22]. We regarded the latter as the more severe pneumonia pattern. Fever was defined as >38.0 °C as measured via ear-drum thermometry. The first fever day was regarded as the first day of illness. Hospitalization day was calculated as discharge date minus admission day plus one. Intractable cases were defined as having a fever duration of >5 days and/or progressive pneumonia after treatment with initial corticosteroids. We analyzed demographic, clinical, chest radiographic, and laboratory findings through review of medical records, and compared effects among different treatment groups.

### 2.3. Ethics

Written informed consent was obtained from caregivers of all children to allow their clinical records to be used in this study. The study was conducted in accordance with the Declaration of Helsinki and was approved by the Institutional Review Board of The Catholic University of Korea, Daejeon St Mary′s Hospital (IRB number: DC18RESI0102).

### 2.4. Statistical Analysis

All calculations were performed with SPSS ver. 14.0 (SPSS Inc., Chicago, IL, USA). The data were expressed as mean ± standard deviation (SD) or median (min–max) for continuous variables or as number of cases (percentage) of a specific group for categorical variables. Comparisons between groups were performed by Mann-Whitney test or paired t-test (Wilcoxon) for continuous variables, and chi-square or Fisher’s exact tests for categorical variables. All *p*-values were two-tailed, and *p*-values of <0.05 were considered statistically significant.

## 3. Results

### 3.1. Demographic, Clinical, and Laboratory Findings of MP Pneumonia Patients

The subject included a total of 257 patients. The mean age of the patients was 5.6 ± 3.1 years of age (range: five months to 15 years), and the age distribution of the subjects is presented in Figure 1. The male-to-female ratio was 1:1 (130:127). In IgM titration tests performed twice during the period of hospital admission, 69% of patients showed increased IgM antibody titers and 13% of patients were seroconverters (from negative to various positive titers), and 18% of patients showed titers of ≥1:640 in both tests. Mean hospitalization, fever duration prior to admission, and total fever duration were 6.0 ± 1.8 days, 5.1 ± 2.6 days, and 5.6 ± 2.8 days, respectively.

Initially, 114 (44%) patients received corticosteroid treatment with oral prednisone (1 mg/kg), while 100 (39%) patients received low-dose IVMP (1–2 mg/kg) and 43 (17%) patients received high-dose IVMP (5–10 mg/kg). All patients received steroids within 36 h of presentation, and 234 patients (91%) received steroids within 24h after admission. After steroid administration, 190 patients (74%) experienced immediate defervescence within 24 h (the next day), 235 patients (91%) had fevers that subsided within 48 h, and 246 patients (96%) had fevers that subsided within 72 h after admission. Eight patients (3%) with persistent fever and/or disease progression for 36–48 h were treated with additive high-dose IVMP (5 mg/kg or 10 mg/kg): initially five patients received oral prednisolone and three patients received low-dose IVMP. All patients treated with additive steroids responded well to treatment, and no patient had a fever for over 48 h or disease progression after high-dose steroid treatment. There were no patients who were treated in the intensive care unit, and there were no adverse reactions following steroid treatment. With respect to antibiotic treatment, 172 patients (67%) were treated with a broad-spectrum antibiotic and clarithromycin, and 85 patients were treated with only a broad-spectrum antibiotic. There was only one patient as an intractable case, but total fever duration after admission was eight days without pneumonia progression. Sixty-eight patients (26%) had bronchopneumonia and 189 patients had segmental/lobar pneumonia (Table 1).

The laboratory findings of patients in this series, including the white blood cell (WBC) count with differentials and C-reactive protein (CRP), are shown in Table 2. These findings were similar to those we observed in previous studies [21,22].

### 3.2. Comparison of Patients Treated With Macrolide and Those Without

We compared clinical and laboratory parameters of 172 children treated with clarithromycin, and 85 treated without clarithromycin. There were no significant differences in clinical parameters between groups, including age, fever duration, steroid treatments, pneumonia pattern, and hospitalization (Table 1). There were no significant differences in laboratory parameters including WBC count, CRP, and lactate dehydrogenase (LDH) values (Table 2).

## 4. Discussion

In the present study, we demonstrated that the majority of patients experienced rapid defervescence and improved clinical symptoms within 24–48 h after early corticosteroid treatment irrespective of used antibiotics, and no patients progressed to intractable or refractory cases after treatment with a dose-adjusted corticosteroid therapy based on clinical severity. Although it is a reasonable hypothesis that macrolide antibiotics are less effective in epidemics involving MRMP strains, we used clarithromycin for MP pneumonia patients in the recent epidemic in which MRMP strains might be prevalent. We observed that there were no differences in total fever duration and rate of sustained fever after initial steroid treatment between patients treated with a beta-lactam only and those treated with additional clarithromycin. These findings suggest that antibiotics have limited effects on MP pneumonia. The immunopathogenesis of MP pneumonia may be associated with a hyper-immune reaction of the host and is not associated with pathogen-induced cytopathy [11].

Although MP pneumonia is a self-limiting disease, antibiotics have been recommended for treating MP pneumonia patients [24,25]. However, it was overlooked that antibiotic treatment could not prevent from the progression of disease from pharyngitis to pneumonia and/or pneumonia progression in some severely affected patients [5,6,26]. Despite the early reports, pediatricians have experienced that some patients with MP pneumonia do not respond to antibiotics [7,8,9]. There have been few well-designed and controlled studies for antibiotic’s effect on MP infection such as antibiotic treated group vs. no treated group. Outcome studies of patients treated with empiric antibiotic coverage for atypical pathogens have not shown clinical efficacy in hospitalized children and adults with community-acquired pneumonia [12,27]. Although alternative antibiotics, such as tetracycline and quinolones, have been reported to induce more rapid defervescence in children with MRMP infections, outcomes such as absence of fatality or severe morbidity did not differ between treatment groups [28,29], suggesting that MP pneumonia is a self-limiting disease. Additionally, some patients treated with alternative antibiotics might tend to be treated with corticosteroids [29]. Some study groups reported that there were no clinical differences between patients with MRMP and MSMP [30], and that early macrolide treatment or macrolide resistance did not contribute to the clinical severity of MRMP pneumonia [31,32]. On the other hand, the use of alternative antibiotics in children is limited at present, due to complications such as tooth discoloration and injury to growing joint cells [33].

The immunopathogenesis of MP infection and the role of immune modulators for MP pneumonia are not fully understood. Antibiotics are well-known to have a limited effect on the natural course of acute viral infections, acute infection-related immune mediated diseases, including acute rheumatic fever and acute streptococcal glomerulonephritis caused by group A beta-streptococci, and Kawasaki disease, which is associated with substances produced after exposure to unknown pathogen(s) [34,35]. Although MP is classified as a small extracellular bacterial species microbiologically [36], it is proposed that MP may act like a virus in the pathogenesis of the disease [1,11]. Together with clinical and epidemiological similarities to viral respiratory infections, in vitro studies reported that some MP species can invade into host cells like viruses or intracellular bacteria such as the Chlamydiae and Legionellae species [37]. Recently, it was reported that *Mycoplasma agalactiae* could enter host cells and disseminate systemically to distant organ cells in a sheep model [38]. Thus, it is possible that inflammation-inducing substances in MP infection are produced when pathogens are replicated within host cells. It is proposed that the host immune system controls these etiologic substances that originate from pathogens, including toxins and pathogen associated molecular patterns (PAMPs), and/or those originated from injured host cells including damage associated molecular patterns (DAMPs), pathogenic proteins, and pathogenic peptides [39,40]. When these substances spread systemically and locally and bind to target organ cells, clinical symptoms begin due to the activation of corresponding immune cells and immune proteins. The substances produced from injured host cells induce further inflammation if released into the systemic circulation or near local lesions. Therefore, early control of lung injuries from initial hyperactive immune reactions that may be performed by non-specific adaptive immune cells is crucial for reduction of morbidity and prevention of pneumonia progression in patients with MP pneumonia as well as other types of pneumonia including severe influenza pneumonia [40,41,42]. Besides many reports regarding beneficial effect of corticosteroids in severe MP pneumonia, it has been reported that early additional corticosteroid therapy for severe adult patients with community acquired pneumonia is helpful for reducing morbidity and treatment failures [43,44].

In the present study, we used corticosteroids to treat patients at earlier disease stages than in previous studies, with the majority of our patients (91%) receiving treatment within 24 h after admission. The total fever duration and hospitalization in this series (mean 5.6 days and 6.0 days, respectively) were shorter than those observed during a 2011 epidemic (6.3 days and 6.4 days, respectively) [22]. All patients treated during the 2015–2016 epidemic received corticosteroids within 24–36 h, while during the 2011 epidemic only half of our patients, those with fever duration of ≥48 h after hospitalization, received corticosteroids. This finding also suggests that the early control of initial pneumonia is essential for the reduction of morbidity and prevention of disease progression. The total fever duration in 2015–2016 series and in the 2011 epidemic were far shorter than those in MRMP infected patients treated with macrolides or late corticosteroid-treated patients in Japan and in Korea (8–12 days) [17,18,19,45,46]. Since the severity of immune reaction varies in individuals with MP pneumonia and corticosteroid effect is dose-dependent, some severely affected patients may need higher doses for initial treatment. In this study, 43 (17%) patients who had severe clinical manifestations at presentation, and eight (3%) patients who had persistent fever or disease progression after initial steroids received high-dose IVMP (5 mg/kg/day or 10 mg/kg/day). There were no patients who had a fever duration of ≥2 days after 10 mg/kg of IVMP infusion including one intractable case. Corticosteroid dose could be determined on a case-by-case basis, as shown in this series and in our previous studies. Long-term use of high-dose corticosteroids can induce immune suppression and other detrimental complications. Our treatment policy composed of high-dose, short-term, and rapid-tapering of corticosteroids may reduce such risks [22].

There are some limitations in this study. This study was not a comparative study for efficacy of corticosteroids because of no control group. We used antibiotics for all subjects, but the exact role of antibiotics, including beta lactams, on MP infection could not be determined because there was no control group without antibiotic treatment. We were unable to confirm MRMP strains, but other investigators in Korea have reported that the strains in the recent epidemic were MRMP strains [16,17,46]. We used corticosteroids prior to the definitive diagnosis of MP pneumonia. However, patients were selected in the MP epidemic period and there were few patients who had contraindications for corticosteroid use. In our experience high-dose, short-term, and rapid-tapering corticosteroid treatment was effective not only in severe MP pneumonia but also in viral infections, including severe influenza pneumonia [42,47] and severe acute bronchiolitis without adverse reactions [40].

## 5. Conclusions

Antibiotics may have limited effects on MP infection, since no progressive cases were noted among MRMP pneumonia patients treated with a macrolide or those without macrolide. Although MP pneumonia is a self-limiting disease, early corticosteroid treatment may hasten disease recovery and prevent disease progression. Well-controlled studies on the roles of corticosteroids and antibiotics in the treatment of MP pneumonia are needed.

## Figures and Tables

**Figure 1 jcm-08-00726-f001:**
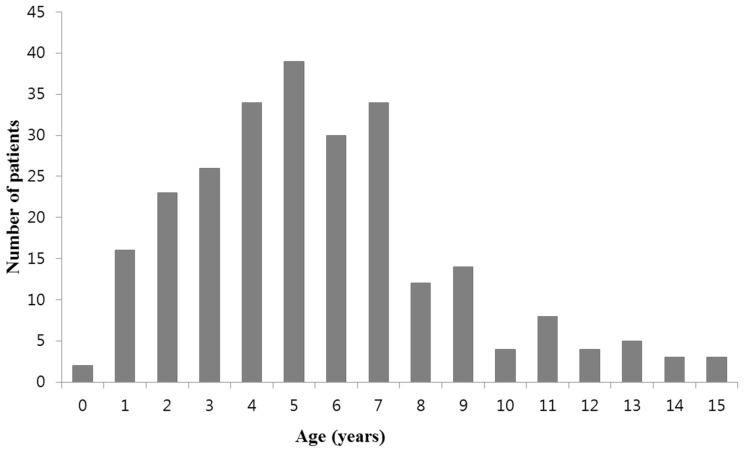
Age distribution of patients in this study.

**Table 1 jcm-08-00726-t001:** Clinical characteristics in all patients and comparison between the patients treated with macrolide and without.

Clinical Characteristics	All	Macrolide (+)	Macrolide (−)	*p*-Value *
	(*n* = 257)	(*n* = 172)	(*n* = 85)	
Age (year)	5.6 ± 3.1	5.7 ± 3.5	5.4 ± 2.9	0.614
Male-female ratio	130:127	86:86	44:41	0.793
Diagnosis (*n*, %) †				
Increased titers	177 (69)	121 (70)	56 (66)	0.477
Seroconversion	33 (13)	15 (9)	18 (21)	0.06
High titers ≥1:640	47 (18)	36 (21)	11 (13)	0.127
Hospitalization (day)	6.0 ± 1.8	6.1 ± 1.9	5.9 ± 1.4	0.424
Duration of fever (day)				
Before admission	5.1 ± 2.6	5.1 ± 2.5	5.2 ± 2.9	0.683
Total duration	5.6 ± 2.8	5.7 ± 2.8	5.5 ± 2.9	0.621
Corticosteroids, *n* (%)				
Oral prednisolone (1 mg/kg)	114 (44)	80 (47)	34 (40)	0.352
Intravenous MP (1–2 mg/kg)	100 (39)	62 (36)	38 (45)	0.221
High-dose MP (5 mg/kg or 10 mg/kg)	43 (17)	30 (17)	13 (15)	0.619
Additive MP (5 mg/kg or 10 mg/kg)	8 (3)	6 (4)	2 (2)	0.724
Pneumonic infiltration, *n* (%)				
Bronchopneumonia	68 (26)	47 (27)	21 (25)	0.764
Segmental/lobar pneumonia	189 (74)	125 (73)	63 (75)	0.882

* Statistical analysis was performed between the group with macrolide and the group without macrolide. Continuous variables are expressed as mean ± standard deviation and categorical variables are expressed as case′s number (%). † Diagnosis were made on seroconversion (negative to positive), increased titer (four-fold or greater increased) or high tier (≥1:640) in paired examinations. MP, methylprednisolone.

**Table 2 jcm-08-00726-t002:** Comparison of laboratory findings between the patients treated with macrolide and without.

Laboratory Parameters	All	Macrolide (+)	Macrolide (−)	*p*-Value
	(*n* = 257)	(*n* = 172)	(*n* = 85)	
WBC (×10^3^/µL)	8.2 (1.5–28.5)	7.9 (1.5–28.5)	8.3 (4.1–25.4)	0.202
Neutrophil (%)	62.9 (16.0–88.5)	62.4 (18.0–83.9)	63.6 (16.0–88.5)	0.786
Lymphocyte (%)	25.7 (6.4–75.3)	26.0 (7.0–65.2)	24.8 (6.4–75.3)	0.930
Monocyte (%)	8.1 (0.2–22)	8.2 (0.2–17)	8.0 (0.7–22)	0.731
Hemoglobin (g/dL)	12.1 (10.1–15.5)	12.1 (10.1–15.5)	12.2 (10.6–14.6)	0.443
ESR (mm/h)	24 (3–84)	25 (3–84)	23 (3–72)	0.145
CRP (mg/dL)	2.4 (0.1–19.9)	2.4 (0.1–14.3)	2.4 (0.1–19.9)	0.544
LDH (IU/L)	287 (23–1600)	295 (23–748)	283 (147–1600)	0.174
ALP (IU/L)	159 (58–323)	157 (58–293)	165 (64–323)	0.131
AST (IU/L)	30 (15–908)	30 (15–299)	29 (16–908)	0.855
ALT (IU/L)	14 (3–1638)	14 (3–295)	14 (6–1638)	0.572

Continuous variables are expressed as medians (min – max). WBC, white blood cell; ESR, erythrocyte sedimentation rate; CRP, C-reactive protein; LDH, lactic dehydrogenase; ALP, alkaline phosphatase; AST, aspartate aminotransferase; and ALT, alanine aminotransferase.

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
