# Peer review of "Early Corticosteroid Therapy for Mycoplasma pneumoniae Pneumonia Irrespective of Used Antibiotics in Children"

_jcm, 2019, doi:10.3390/jcm8050726_

Round 1
Reviewer 1 Report
The authors have made an effort to address my concerns. Although I still think that lack of control groups is a major drawback, I understand that the information provided in this article could be of interest for pediatricians treating children infected with MP.
The authors ackowledged the limitations of the study, which could help to design further studies on this issue.
Steroids treatment seems to be safe and could represent an alternative to treat MP.
Reviewer 2 Report
Dear Editor, thanks for the invitation. The authors respond all my comments.
This manuscript is a resubmission of an earlier submission. The following is a list of the peer review reports and author responses from that submission.
Round 1
Reviewer 1 Report
The authors show here their results using steroids in a cohort of patients with Mycoplasma pneumoniae pneumonia. They conclude that early corticosteroid therapy might reduce disease morbidity and prevent disease progression in MP pneumonia patients without side effects, and antibiotics may have limited effects on MP infection.
This reviewer has a number of serious concerns regarding this submission:
- ALL the patients received steroids and antibiotics….since this is an uncontrolled study, the potential effect of receiving no antibiotics and no steroids is not known….The authors cannot support they conclusions, since they lack of control groups (in fact the authors acknowledge in the discussion that “This study was not a comparative study for efficacy of corticosteroids because of no control group.”) They only can say that steroid administration seems to be safe in this context, and that addition of macrolides to the antibiotic treatment seems to provide no additional benefit.
- This affects the discussion. The authors state that “We observed that there were no differences in total fever duration and rate of sustained fever after initial steroid treatment between patients treated with a beta-lactam only and those treated with additional clarithromycin. These findings suggest that antibiotics have limited effects on MP pneumonia”. How can you suggest this in the absence of a control group receiving no antibiotics?
- This conclusion is thus not supported by the findings “In conclusions, antibiotics may have limited effects on MP infection, since no progressive cases were noted among MRMP pneumonia patients treated with a macrolide or those without macrolide”. The results support the absence of benefit of adding macrolides, NOT of avoiding antibiotics, since ALL the patients were treated with antibiotics.
-Have the authors considered using hystorical control cohorts?
The article should be reoriented to communicate the safety of steroids use in MP, and the safety of no adding macrolides to the antibiotic treatment in this disease.
Minor comments:
Abstract: some info is missing here: “ The mean age and the male:female ratio were 5.6 ± 3.1 years, respectively.”
Discussion: please correct “defervscence”
Responses to the commentsof Reviewer 1
1) The authors show here their results using steroids in a cohort of patients with Mycoplasma pneumoniae pneumonia. They conclude that early corticosteroid therapy might reduce disease morbidity and prevent disease progression in MP pneumonia patients without side effects, and antibioticsmay have limited effects on MP infection.
This reviewer has a number of serious concerns regarding this submission:
-ALL the patients received steroids and antibiotics….since this is an uncontrolled study, the potential effect of receiving no antibioticsand no steroids is not known….The authors cannot support they conclusions, since they lack of control groups (in fact the authors acknowledge in the discussion that “This study was not a comparative study for efficacy of corticosteroids because of no control group.”) They only can say that steroid administration seems to be safe in this context, and that addition of macrolides to the antibiotic treatment seems to provide no additional benefit.
-This affects the discussion. The authors state that “We observed that there were no differences in total fever duration and rate of sustained fever after initial steroid treatment between patients treated with a beta-lactam only and those treated with additional clarithromycin. These findings suggest that antibiotics have limited effects on MP pneumonia”. How can you suggest this in the absence of a control group receiving no antibiotics?
R: We thank the reviewer for the comprehensive review of our manuscript. We agree with the reviewer’s comment that since there was no control group without antibiotic treatment, the effect of antibiotics on Mycoplasmapneumoniae(MP) infection could not be determined through this study. However, it is already well-known that MP strains are sensitive only to macrolides, tetracycline, and quinolones; and some study groups have reported that macrolides were effective in a portion of patients infected with macrolide resistant MP (MRMP) strains. Additionally, it is likely that not all subjects included this study were infected with MRMP strains. Thus, our results between patients treated with a beta-lactam only could be interpreted as those without antibiotic treatment effective for MP, and those treated with an additional clarithromycin could be interpreted as no antibiotic treatment effective for MRMP strains.
2) This conclusion is thus not supported by the findings “In conclusions, antibiotics may have limited effects on MP infection, since no progressive cases were noted among MRMP pneumonia patients treated with a macrolide or those without macrolide”. The results support the absence of benefit of adding macrolides, NOT of avoiding antibiotics, since ALL the patients were treated with antibiotics.
R: In general, initial or empirical use of antibiotics for any pneumonia in hospitalized patients has been an acceptable practice because of unknown etiological agents at initial presentation. Although MP is sensitive to macrolides, we believed that other antibiotics, including beta-lactams, may have influence on the clinical course of MP infection because some patientswith severe MP pneumonia or viral pneumonia are prone to have secondary bacterial infections. Further, wepreviously postulatedthat antibioticsmay have influence on not only MP but also normal flora that canpossibly be helpful for replication of MP in the host, becauseof fastidious growing nature of MPs in the culture systemwithout feeding cells [Reference 22].
3) Have the authors considered using historical control cohorts?
R: We have reviewed the limitation of antibiotics on MP infection in the Discussion section, referencing data on other studies.
4) The article should be reoriented to communicate the safety of steroids use in MP, and the safety of no adding macrolides to the antibiotic treatment in this disease.
R: For these issues, we added sentences in the limitation paragraphs in Discussion section. This study may be informative for pediatricians who contemplate whether to use alternative antibiotics or add corticosteroids in antibiotic-nonresponsive severe cases in the era of MRMP strains. Indeed, many pediatricians in Korea have experienced no treatment failures by using a macrolide with corticosteroids in the recent epidemics. We hope for your kind understanding, and accept the main point of this study.
5)Minorcomments
-Abstract: some info is missing here: “ The mean age and the male:female ratio were 5.6 ± 3.1 years, respectively.”
-Discussion: please correct “defervscence”
R: We have corrected the abstract and the discussion as the reviewer has mentioned.
2
with macrolide resistant MP (MRMP) strains. Additionally, it is likely that not all subjects included this study were infected with MRMP strains. Thus, our results between patients treated with a beta-lactam only could be interpreted as those without antibiotic treatment effective for MP, and those treated with an additional clarithromycin could be interpreted as no antibiotic treatment effective for MRMP strains.
2) This conclusion is thus not supported by the findings “In conclusions, antibiotics may have limited effects on MP infection, since no progressive cases were noted among MRMP pneumonia patients treated with a macrolide or those without macrolide”. The results support the absence of benefit of adding macrolides, NOT of avoiding antibiotics, since ALL the patients were treated with antibiotics.
R: In general, initial or empirical use of antibiotics for any pneumonia in hospitalized patients has been an acceptable practice because of unknown etiological agents at initial presentation. Although MP is sensitive to macrolides, we believed that other antibiotics, including beta-lactams, may have influence on the clinical course of MP infection because some patientswith severe MP pneumonia or viral pneumonia are prone to have secondary bacterial infections. Further, wepreviously postulatedthat antibioticsmay have influence on not only MP but also normal flora that canpossibly be helpful for replication of MP in the host, becauseof fastidious growing nature of MPs in the culture systemwithout feeding cells [Reference 22].
3) Have the authors considered using historical control cohorts?
R: We have reviewed the limitation of antibiotics on MP infection in the Discussion section, referencing data on other studies.
4) The article should be reoriented to communicate the safety of steroids use in MP, and the safety of no adding macrolides to the antibiotic treatment in this disease.
R: For these issues, we added sentences in the limitation paragraphs in Discussion section. This study may be informative for pediatricians who contemplate whether to use alternative antibiotics or add corticosteroids in antibiotic-nonresponsive severe cases in the era of MRMP strains. Indeed, many pediatricians in Korea have experienced no treatment failures by using a macrolide with corticosteroids in the recent epidemics. We hope for your kind understanding, and accept the main point of this study.
5)Minorcomments
-Abstract: some info is missing here: “ The mean age and the male:female ratio were 5.6 ± 3.1 years, respectively.”
-Discussion: please correct “defervscence”
R: We have corrected the abstract and the discussion as the reviewer has mentioned.
Reviewer 2 Report
Dear Editor, i read with interest the article by Yang et al. about early corticosteroid therapy for Mycoplasma pneumoniae pneumonia in children. The authors reported an interesting and necessary that early corticosteroid treatment in MP pneumonia in children might reduce disease morbidity and prevent disease progression, whereas the authors suggested that antibiotics may have limited effects on MP infection. The main limitation is the retrospective nature of the study and the fact that corticosteroids were used before MP diagnosis. However, as the authors addressed in limitation section they select patients in MP epidemic period. I have minor comments about the study:
1. - abstract, line 17, please include the percentage of “eighty-five patients” received only ……. Or report data in the same format that line 20 (246/257). It will be helpful to report entire percentages, i.e, instead 95.7% put 98% . Also report percentages in line 21 “eight patients who received….” Also report percentages in line 23 when compared clinical or laboratory differences.
2. - introduction, please include the aim of the study in the last paragraph of the introduction and the main outcomes to investigate.
3. - methods, please include section of definition, and include i.e pneumonia definition, MP pneumonia, etc
4. - results, please reported percentages without fraction, line 133 74.0%% and correct the mistake
5.- tables, I recommend report the percentages not with fraction
Response to the comments of Reviewer 2:
1) Dear Editor, i read with interest the article by Yang et al. about early corticosteroid therapy for Mycoplasma pneumoniae pneumonia in children. The authors reported an interesting and necessary that early corticosteroid treatment in MP pneumonia in children might reduce disease morbidity and prevent disease progression, whereas the authors suggested that antibiotics may have limited effects on MP infection. The main limitation is the retrospective nature of the study and the fact that corticosteroids were used before MP diagnosis. However, as the authors addressed in limitation section they select patients in MP epidemic period. I have minor comments about the study:
R: We appreciate the reviewer’s informative and comprehensive review of our manuscript.
2) Abstract, line 17, please include the percentage of “eighty-five patients” received only ……. Or report data in the same format that line 20 (246/257). It will be helpful to report entire percentages, i.e, instead 95.7% put 98%. Also report percentages in line 21 “eight patients who received….” Also report percentages in line 23 when compared clinical or laboratory differences.
R: We have corrected the abstract as the reviewer has recommended.
3) Introduction, please include the aim of the study in the last paragraph of the introduction and the main outcomes to investigate.
R: We added the aim of the study and the main outcomes to investigate in the introduction section.
4) Methods, please include section of definition, and include i.e pneumonia definition, MP pneumonia, etc.
R: We added the definition for pneumonia in the methods section as advised.
5) Results, please reported percentages without fraction, line 133 74.0%% and correct the mistake.
R: Changes were made to the Results as recommended.
6) Tables, I recommend report the percentages not with fraction
R: We changed to percentages and deleted the fractions as advised.